# Metallothionein Expression and its Influence on the In Vitro Biological Behavior of Mucoepidermoid Carcinoma

**DOI:** 10.3390/cells9010157

**Published:** 2020-01-08

**Authors:** João Rafael Habib Souza Aquime, Lara Carolina D’Araújo Pinto Zampieri, Maria Sueli da Silva Kataoka, Nelson Antonio Bailão Ribeiro, Ruy Gastaldoni Jaeger, Artur Luiz da Silva, Rommel Thiago Jucá Ramos, Sérgio de Melo Alves Júnior, João de Jesus Viana Pinheiro

**Affiliations:** 1Department of Oral Pathology, School of Dentistry, Federal University of Para, Avenida Augusto Correa, 01. Belem, Para 66075-110, Belém, PA 66073-000, Brazil; joao_habib@hotmail.com (J.R.H.S.A.); lara.krol@gmail.com (L.C.D.P.Z.); sukataoka@yahoo.com.br (M.S.d.S.K.); sergiomalves@gmail.com (S.d.M.A.J.); 2Laboratory of Genetics and Molecular Biology, State University of Pará, Travessa Perebebuí 2623, Belém, PA 66087-670, Brazil; nelsonribeiro07@yahoo.com.br; 3Nucleus of Genetic Analysis for Imaging, Innovations Technologic Centre, Institute Evandro Chagas, Rodovia BR-316 km 7 s/n, Ananindeua, PA 67030-000, Brazil; 4Department of Cell and Developmental Biology, Ed. Biomédicas 1, Sala 410, Institute of Biomedical Sciences, University of São Paulo, Av. Prof. Lineu Prestes 1524, São Paulo, SP 05508-000, Brazil; rgjaeger@usp.br; 5Institute of Biological Sciences, Federal University of Pará, Augusto Corrêa Avenue 01, Belém, PA 66075-110, Brazil; arturluizdasilva@gmail.com (A.L.d.S.); rommelthiago@gmail.com (R.T.J.R.)

**Keywords:** mucoepidermoid carcinoma, metallothionein, matrix metalloproteinases, extracellular matrix, salivary glands

## Abstract

Mucoepidermoid carcinoma (MEC) is the most common tumor in the salivary glands, often presenting with recurrence and metastasis due to its high invasive capacity. Metallothionein (MT), a zinc storage protein that supplies this element for protease activity, is probably related to mucoepidermoid carcinoma behavior. This prompted us to characterize a cell line derived from mucoepidermoid carcinoma and to correlate metallothionein expression with transforming growth factor-α (TGF-α), tumor necrosis factor-α (TNF-α) and matrix metalloproteinases (MMPs). Transcriptomic analysis and cytogenetic assays were performed to detect the expression of genes of interest and cellular chromosomal alterations, respectively. MEC cells with a depleted metallothionein 2A (*MT2A*) gene were subjected to Western blot to correlate metallothionein expression with growth factors and MMPs. Additionally, cells with depleted MT were subjected to migration and invasion assays. The transcriptomic study revealed reads mapped to cytokeratins 19 and AE1/AE3, α-smooth muscle actin, vimentin, and fibronectin. Cytogenetic evaluation demonstrated structural and numerical alterations, including the translocation t(11;19)(q21;p13), characteristic of MEC. Metallothionein depletion was correlated with the decreased expression of TGF-α and MMP-9, while TNF-α protein levels were augmented. Migration and invasion activity were diminished after metallothionein silencing. Our findings suggest an important role of MT in MEC invasion, through the regulation of proteins involved in this process.

## 1. Introduction

Mucoepidermoid carcinoma (MEC) is the most common tumor in the salivary glands, representing about 30% of all salivary gland malignancies [1]. Its clinical presentation is frequently as a slow-growing tumor without associated pain, mainly located on the parotid gland. MEC has significant rates of recurrence and metastasis; its behavior being related to the histological grade of the tumor, which has been confirmed by retrospective studies [2], and to cellular differentiation, cystic spaces and cytologic atypia [3]. Histologically, MEC is composed of three cell populations: mucous, intermediate and epidermoid. The predominant cellular subtype is directly linked with the gradation of the tumor into low, intermediate or high grades of malignancy [3].

Mucoepidermoid carcinoma (MEC)’s local invasiveness is probably responsible for tumor recurrence and metastasis. The mechanism of invasiveness in salivary gland tumors has already been demonstrated in some studies and is related to proteolysis of the extracellular matrix (ECM), migration and cell invasion [4]. The proteolytic activity of the matrix, which promotes a physical space for the cells to reach deeper tissues, has been attributed to a family of zinc-dependent enzymes secreted by some cells, called matrix metalloproteinases (MMPs). Possibly, ECM degradation releases growth factors (GF) that activate cell signaling pathways, resulting in increased proliferative activity of the cell line. Growth factors also promote the secretion of MMPs, providing a positive feedback mechanism favorable to tumor invasion [5]. These events may occur with the contribution of metallothionein (MT), a low molecular weight and intracellular protein related to zinc storage [6]. Metallothionein is important for protein and nucleic acid synthesis, elevating metabolic activity and cell proliferation [7]. Thus, MT overexpression has been repeatedly related to a poor prognosis in tumors of the lung, pancreas, prostate and oral cavity [8]. Patients with squamous cell carcinoma with high MT expression have significantly shorter survival rates compared to those with a lower expression of this protein [9].

The relevance of metallothionein in epidermoid carcinoma prognosis has already been established [9]. However, little is known about its role in MEC invasiveness. Considering the histopathological similarities between both tumor types, it is possible that MT is also present and participates in the tumorigenesis of MEC. Moreover, this protein has been linked to the protection of tumor cells against chemotherapy and radiation, which develop a neoplastic resistance to these treatments [10]. Recent studies confirmed that the presence of different MT isoforms, mostly metallothionein 2A (MT-2A), correlates with cell proliferation and clinicopathological behavior in some cancers [11,12]. Furthermore, MT-2A has been considered a relevant prognostic marker in salivary gland tumors, associated with higher invasive activity of the neoplasms [12].

The invasiveness of MEC is probably a complex process mediated by both cell proliferation and ECM proteolysis, among other factors. The impulse to proliferate occurs through the action of growth factors, such as TGF-α (transforming growth factor-α), which is biosynthesized in tumor cells of carcinomas of the salivary glands, possibly acting as a pro-mitotic agent for cell proliferation [13]. However, it is known that tumor development involves not only stimulatory factors, but also inhibitory ones [14], which aim to contain the neoplastic growth. Tumor necrosis factor-α (TNF-α), a cytokine regulator of apoptosis, is an example of such an inhibitor. It has already been demonstrated that TNF-α exhibits low expression in fibroblast and tumor cells [15]. This could explain TNF-α’s role, acting in a manner contrary to tumor development [15]. On the other hand, metallothionein appears to be pro-mitotic and anti-apoptotic [16]. Thus, it is important to study the relationship between TNF-α and metallothionein.

MMPs are proteases responsible for tissue remodelling, especially the degradation of ECM components, including collagens, elastins, gelatin and proteoglycans [17], thus allowing the advancement of tumor cells into the bloodstream. For this reason, they are considered essential for tumor invasion and metastasis in various neoplasms. Among them, the gelatinases MMP-2 and -9 have an important role in degrading type IV collagen, one of the main components of the ECM [4]. Studies have shown that the expression of MMP-2 and -9 is linked with the biological behavior of salivary gland tumors [18].

In this paper, we characterized a cell line derived from mucoepidermoid carcinoma and correlated the expression of metallothionein with transforming growth factor-α (TGF-α), tumor necrosis factor-α (TNF-α) and matrix metalloproteinases (MMPs).

## 2. Materials and Methods

### 2.1. Samples

This study was approved by the Ethics Committee of the Institute of Health Sciences, Federal University of Pará (no. 358.227). A cell line, derived from the primary culture of an MEC, was cultured in Dulbecco’s Modified Eagle’s Medium/Nutrient Mixture F-12 (DMEM-F12; Sigma Chemical Co., St. Louis, MO, USA) supplemented by 10% fetal bovine serum (Gibco, Carlsbad, CA, USA), 2 mM glutamine (Sigma^®^), 3 mM sodium bicarbonate (Sigma^®^), glucose (33 mM; Merck AS, Taquara, RJ, Brazil), 100 UI/mL penicillin (Gibco), 100 μg/mL streptomycin (Gibco^®^) and 2.5 μg/mL Fungizone (Gibco^®^) solution. Cells were kept in a humidified atmosphere of 5% CO_2_ at 37 °C.

### 2.2. Conventional Cytogenetic Analysis

Chromosome metaphases were determined from a cell culture of flasks with an area of 25 cm^2^, to which 0.1 mL of colchicine was added at a concentration of 0.0016% for 1 h. Afterwards, the material was transferred to a centrifuge tube and subjected to a hypotonic process with KCl solution (0.56%), and then fixed with Carnoy’s fixative (three parts of methanol to one part of glacial acetic acid). Subsequently, the cell suspension was dropped onto glass slides carefully and left to dry at room temperature. After that, slides were stained with Giemsa solution (Merck SA, RJ, Brazil) and submitted to the G-banding technique [19,20] with Wright solution (Sigma^®^). Finally, processed slides were viewed in an Axiophot photomicroscope (Zeiss^®^), and images were captured using an Axiocam digital camera (Zeiss^®^) coupled to a microscope with an immersion objective of 100× and a 10× eyepiece. For the cell analysis, the BandView software system was utilized (BandView associated with the Case Data Manager (CDM) and Karyotyping software). Chromosomes were then arranged to consider the morphology, in decreasing order of size.

### 2.3. Transcriptomics Analysis

Analysis of the transcribed messenger RNA (mRNA) of two cell lines (human salivary gland (HSG) and MEC) was performed in three basic stages:
Extraction and mRNA capture: we used 5 × 10^4^ cells of each line, separated into different tubes. Each tube was centrifuged at 16,000 rpm for 10 min, and the culture medium discarded. After washing with PBS 1× and further centrifugation, the cells were exposed to lysis buffer (Lysis/Binding Buffer, Life Technologies AS, Oslo) and a bead system (Dynabeads^®^ Oligo (dT) 25, Life Technologies AS, Oslo, Norway) for 5 min at 25 °C with gentle agitation. After the initial capture, the tubes were placed on a magnetic separation rack for 1 min; purifications were performed with wash buffers A and B (Washing Buffer A and Buffer B, Life Technologies AS, Oslo, Norway), following the manufacturer’s instructions. The beads were exposed to another buffer for separation of these strands of mRNA (Dynabeads^®^ mRNA DIRECT™ Micro Kit Life, Carlsbad, Califórnia, EUA). The samples were quantified (Qubit^®^ HS Assay Kit and RNA Qubit^®^ 2.0 Fluorometer Kit, Life Technologies, Carlsbad, Califórnia, EUA) and stored at −80 °C;PCR amplification: after mRNA extraction was performed, samples were amplified using the PCR method. For this, newly acquired filaments need to be fragmented into smaller ones that enable transcriptomic analysis (100–150 base pairs) in high-temperature cycles, and enzymatic action. Thereafter, the fragments were annealed and exposed to reverse transcription to form complementary strands of DNA (cDNA). The samples were purified at each stage, following the manufacturer’s directions (Ion Total RNA Seq v2 Kit, Life Technologies Austin, USA). Each cDNA strand was taken for the enzymatic ligation of specific adapters, and immersed in the buffer solution. From this, PCR primers were used to create copies of the filaments in 30- and 40-min cycles at high temperatures (Ion Total RNA Seq Primer v2, Life Technologies Austin, USA). The transcriptomic libraries were purified and quantified again (Qubit^®^ HS Assay Kit and RNA Qubit^®^ 2.0 Fluorometer Kit, Life Technologies). To confirm this process, the samples were loaded onto gel Acrilose 1% bromide and subjected to electrophoresis at 100 V for 10 min. Bands confirmed the size of the filaments after exposure to blue light;Reference genome: we used the reference Homo sapiens, assembly GRCh38.p4, available at NCBI (http://www.ncbi.nlm.nih.gov/). The GenBank file was accessed through the Homo sapiens Artemis program [21] to generate a fasta file containing the nucleoid sequence of all chromosomes, and a GFF annotation file with the chromosomes. The TMAP program (https://github.com/iontorrent/TS/tree/master/Analysis/TMAP) was used to perform the mapping of readings against the reference. At this stage, the fasta files for each chromosome of Homo sapiens were used as a reference and so we carried out a mapping of the readings for each condition against each chromosome using the TMAP default parameters. As a result, for each chromosome and condition, a mapping file was generated in SAM format. The SAM files were converted to the format bam ordered using the Samtools program (http://www.htslib.org/). Analysis of differential gene expression was performed using the Cuffdiff program [22], which uses a annotation reference file as the input, as well as the result of the alignments (BAM format) of the reads against the reference. To identify differentially expressed genes, an individual analysis was performed for each of the chromosome conditions analysed. The Cuffdiff generated a list with the information of all genes, including those differentially expressed, for each chromosome.

### 2.4. Small Interfering RNA

Small interfering RNAs against MT2A (siRNA s226631 and s226679, Life Technologies) were used. This is a Silencer Select (Thermo^®^), chemically modified siRNA, with a proprietary chemical modification that reduces overall off-target effects by up to 90% without compromising potency. MEC cells (2 × 10^5^) were cultured in six-well plates, in DMEM-F12 with 10% FBS and without an antibiotic–antimycotic solution, to 60–80% confluence. Cells were incubated with a complex formed by transfection medium (Optimen, Invitrogen Molecular Probes, Carlsbad, CA, USA), Lipofectamine 2000 transfection reagent (Invitrogen^®^) and 40 nM siRNAs targeting MT2A (Life Technologies, New York, USA), following the manufacturer’s instructions. A 40 nM siRNA scrambled sequence (Santa Cruz proprietary target sequence) was used as control. Transfection efficiency was demonstrated by Western blot.

### 2.5. Western Blot

Cells transfected with siRNA targeting MT2A and the control were lysed in RIPA buffer (150 mM NaCl, 1.0% NP-40, 0.5% deoxycholate, 0.1% SDS, 50 mM Tris pH 8.0) with a protease inhibitor cocktail (Sigma^®^). Samples were electrophoresed in 10% polyacrylamide gradient gels. Proteins were transferred to a Hybond ECL nitrocellulose membrane (Amersham^®^) and blocked in TBS with 0.05% Tween 20 (TBST) with 2.5% non-fat milk. The membrane was probed with antibodies against MT 1 and 2 (Abcam^®^), TGF-α (Santa Cruz^®^), TNF-α (Sigma^®^), MMP-2 (Millipore^®^), MMP-9 (DBS^®^) and β-actin (Sigma^®^). Primary antibodies were detected by HRP-conjugated secondary antibodies (1:10,000) and developed using an ECL substrate (Amersham^®^) to reveal the reaction on radiographic films. To probe different antibodies, membranes were stripped with Restore Western Blot Stripping Buffer (Thermo^®^).

### 2.6. Migration Assay

To investigate if MT-2A has influence on the migration of MEC cells, a 10-well bipartite chamber system (NeuroProbe, Inc, Gaithersburg, USA) was used, associated with a porous polycarbonate membrane. The lower chamber wells were filled with DMEM-F12 (Sigma^®^) and 10% fetal bovine serum (Gibco^®^). Cells with expression of MT, reduced by siRNA (10^5^ cells/well) and resuspended in serum-free medium, were placed in the upper chamber of the membrane. Then, the chambers were incubated for 24 h at 37 °C in a humid atmosphere containing 5% CO_2_. After this period, the upper portion of the membrane was removed and carefully scraped for the removal of non-migrated cells, leaving only the cells that had migrated, located on the underside of the membrane. These cells were fixed in 4% paraformaldehyde and stained with 0.2% crystal violet solution in 20% methanol. The acquisition of images using a digital machine (Axiocam MRc, Zeiss) coupled to a microscope (zoom 500×) allowed counting of the migrated cells. The following were used as control: 1) the cells of a scrambled sequence group associated to DMEM-F12, containing 10% FBS in the lower well; 2) cells not transfected, associated to DMEM-F12, containing 10% FBS in the lower well; 3) cells not transfected with FBS-free DMEM-F12 in the lower well.

### 2.7. Invasion Assay

To evaluate whether MT-2A induces an invasive phenotype in MEC cells, the same 10-well bipartite chamber system (NeuroProbe^®^) was used, associated with a porous polycarbonate membrane, covered by 5 μL of Matrigel (Trevigen Inc., Gaithersburg, MD, USA) at a concentration of 14 μg/mL, a substance which corresponds to the basement membrane in vitro. Lower chamber wells were filled with DMEM-F12 (Sigma^®^) and 10% fetal bovine serum (Gibco^®^). In this experiment, cells with MT expression reduced by siRNA (15 × 10^4^ cells/well) were resuspended in a serum-free medium and placed in the upper chamber on the Matrigel-covered membrane. The chamber was incubated for 48 h at 37 °C in a humid atmosphere containing 5% CO2, making Matrigel digestion possible, and the invasion of cells from the upper to lower chamber. After this period, the upper portion of the membrane was removed and delicately scraped to remove non-invaded cells and the Matrigel remains. Cells located in the lower portion were fixed in 4% paraformaldehyde in PBS and stained with 0.2% crystal violet solution in 20% methanol. The acquisition of images using a digital machine (AxiocamMRc, Zeiss) coupled to a microscope (zoom 500×) allowed counting of the invasive cells. The following were used as controls: 1) the cells of a scrambled sequence group associated to DMEM-F12, containing 10% FBS in the lower well; 2) cells not transfected, associated to DMEM-F12, containing 10% FBS in the lower well; 3) cells not transfected, with FBS-free DMEM-F12 in the lower well.

## 3. Results

### 3.1. MEC Cell Line Expresses Epithelial and Mesenchymal Markers

Cytokeratins presented reads mapped in the MEC cell line, mainly CK-7. A higher proportion of reads mapped to α-smooth muscle actin, vimentin and fibronectin were detected, suggesting mesenchymal cells’ prevalence (Table 1).

### 3.2. MT2A and MMP2 Genes are Overexpressed in the MEC Cell Line

Transcriptomic analysis revealed that the *MT2A* gene showed a significantly higher number of reads mapped in the MEC cell line (3789) compared to the HSG line (315). These results reinforce the pronounced expression of MT2A in tumor cells as a relevant prognostic marker. The *MMP2* gene also showed a statistically larger number of reads mapped in MEC (63687) compared to HSG (294) cells (Table 1).

### 3.3. TNFA and MMP9 Genes are Poorly Expressed in MEC

The *TNFA* gene failed to present reads mapped in the MEC cell line. MMP9 expressed only two reads mapped in MEC, suggesting discreet participation of the homonymous proteins encoded by these genes (Table 1).

### 3.4. Conventional Cytogenetic Analysis shows Numerical and Structural Abnormalities

A total of 38 metaphases were analysed, and various alterations were observed. Among the numerical changes verified were: nullisomy in chromosome 15; monosomy in chromosomes 1, 2, 3, 5, 6, 7, 13, 15, 16, 17, 19, 21, 22 and X; trisomy in chromosomes 11, 12, 20 and 21; and tetrasomy in chromosomes 11, 12, 18 and 20. Some of these are described in Figure 1A. Structural alterations, such as deletion of the long arm of one chromosome in pair 4, and the centric fission of a chromosome in pair 1, were detected. The translocation t(11;19) (q21;p13), characteristic of MEC, was also present (Figure 1B).

### 3.5. MT2A Silencing Decreases Expression of TGF-α and MMP-9 and Increases TNF-α Expression in MEC Cells

Western blot demonstrated expression of the proteins of interest, and confirmed MT2A’s silencing efficiency. MEC cells treated with 40 nM of siRNA to the MT2A gene showed decreased expression of MT-2A protein compared to the scrambled siRNA control (Figure 2A). Cells with a depleted MT2A gene promoted a reduction in TGF-α expression (Figure 2B), while augmenting TNF-α protein levels (Figure 2C).

With regards to MMPs, it was found that MMP-2 expression was unaltered by the depletion of MT2A (Figure 2D). On the other hand, both MMP-9 and metallothionein exhibited a decrease in protein levels (Figure 2E). β-actin served as a loading control (Figure 2F).

### 3.6. MT2A Silencing Decreases Migratory and Invasive Activity in MEC Cells

MEC cells with reduced expression of MT2A exhibited a significant decrease in both migration and invasion compared to controls (Figure 3 and Figure 4).

## 4. Discussion

Our findings suggest that metallothionein plays an important role in the tumor invasion mechanism in mucoepidermoid carcinoma, through the regulation of proteins directly involved in this process, such as TGF-α, TNF-α and MMP-9. Moreover, metallothionein also influences both the migratory and invasive activity of the mucoepidermoid carcinoma cell line (MEC). These are novel findings related to the behavior of an important salivary gland tumor.

Mucoepidermoid carcinoma is a significant disease, mainly because of its notable prevalence among salivary gland tumors and its potential for aggressive behavior, with high rates of recurrence and metastasis [1,2]. The development of tumor cell lines has been widely accepted as a model to understand the biological behavior of different neoplasms in vitro. In our paper, we used a cell line from a human MEC, characterized by the expression of cellular proteins, epithelial and mesenchymal markers, and cytogenetic analysis. Cytokeratins (CK) characterize cells of epithelial origin, and their expression, varies with cell type, differentiation degree and level of tissue development. Even after the transformation of normal cells in cancer cells, patterns of CK are maintained, and therefore they are utilized as important tumor markers [23]. CK-AE1/AE3 identifies CK-1, -2, -3, -4, -5, -6, -7, -8, -10, -13, -14, -15, -16 and -19, some of which have already been demonstrated in different types of carcinoma [24,25]. According to Azevedo et al. [26], in MEC, CK-7, -8, -14 and -19 are expressed by squamous cells, whereas the intermediate and mucosal cell populations mainly express CK-7. Our transcriptomic results showed that CK-7 presented more reads mapped in the MEC cell line, suggesting a neoplasia with a prevalence of intermediate and mucous cells, and, thus, possibly classified as low to intermediate grade [3]. Recent immunohistochemical studies have found CK-19 in cells of glandular origin [27], justifying the expression of this protein in the cell line studied, derived from MEC. Furthermore, the higher expression of vimentin and α-smooth muscle actin in the MEC cell line, compared to the HSG line, demonstrates the presence of myoepithelial cells, which are part of cancer-associated fibroblasts (CAFs) and were already visualized in MEC [28]. Moreover, when MT is expressed in CAFs, the tumor invasiveness capacity is increased in uterine cervical carcinoma [29].

The presence of various constitutive molecules of the ECM characterizes its participation not only as a structural component of tissues, but also as a mediator in cell signaling and a regulator in the processes of cell adhesion and migration [14]. In the MEC cell line, we visualized reads mapped to α-smooth muscle actin, vimentin and fibronectin, indicating the presence of mesenchymal and myoepithelial cells. Previous studies had already identified myoepithelial cells in MEC [30,31]. The presence of these cells is identified by the expression of α-smooth muscle actin, considered an excellent tool for detecting these cells in salivary gland tumors [32]. Another marker found, vimentin, is an intermediate filament protein that plays an important role in cytoskeleton regulation, considered a useful marker of mesenchymal cells [33]. Furthermore, vimentin has been proposed as a regulator agent of interaction between proteins of the cytoskeleton and cell adhesion molecules, participating in the processes of adhesion, migration, invasion and signal transduction in tumor cells [34]. Fibronectin, also expressed in the cell line, is considered a major mesenchymal adhesive glycoprotein of the ECM and is responsible for cell–cell and cell–matrix adhesion. For this reason, it has an important influence on cell migration and early differentiation [35].

The proliferative activity of MEC cells probably receives a contribution from the growth factors (GF) present in the cell line. We detected a small number of mapped reads for the gene coding for TGF-α, confirmed by its weak expression in Western blot. The performance of this GF in MEC has been rarely studied, despite its relevance in head and neck cancers, including its role in the development and progression of mucopidermoid carcinoma [36]. In addition, the presence of TGF-α confirms the results found by Gibbons et al. [36], which demonstrated a more intense expression of this GF in MEC than in another highly prevalent tumor of the salivary glands, adenoid cystic carcinoma (ACC).

In the tumorigenesis mechanism, there are proteins with opposing actions to the tumor, which try to contain the spread of the disease and can be decreased or absent in neoplastic progression. Among them, we observed discreet expression by Western blot and no mapped reads for TNF-α, a coherent result as it is a cytokine with the function of promoting apoptosis in tumors [37].

The recurrence of and distant metastasis in MEC are related to the invasive capacity of tumor cells in underlying tissues, making their total elimination more difficult and increasing the risk of persistent cells that can develop, causing the return of the disease. The importance of MMPs and GFs making the mechanisms of tumor invasion possible is well elucidated in the literature. The participation of metallothionein (MT) has also been investigated. This molecule correlates with the supply of zinc for the activation of certain transcription factors, increasing tumors’ proliferative potential. MT also regulates zinc-dependent proteases responsible for ECM degradation [7,10]. Four isoforms have been studied, in which MT-3 and MT-4 are expressed in the brain and in the squamous epithelia, respectively. MT-1 and MT-2 are present in most organs, but their role has not been elucidated [12]. A specific isoform of this protein, MT-2A, has been widely studied as a poor prognostic factor in breast cancer, with its presence associated with more aggressive tumor behavior [12]. Thus, expression of MT in MEC cells, viewed by transcriptomic analysis and Western blot, suggests the probable action of this protein in tumor invasiveness of MEC in vitro.

Corroborating our findings, the presence of MMP-2 and -9 in salivary gland tumors and in MEC has been demonstrated previously and correlated with higher rates of metastasis and disease recurrence, as the function of these proteases is to promote the degradation of various ECM components [4,38,39], thereby increasing the infiltrative capacity of tumor cells.

The identification of specific chromosomal changes in neoplasms is considered another indicator of clinical importance, as the high number of these alterations is correlated with a more advanced tumor grade. Certain chromosomal translocations are characteristic for some types of cancer, and for MEC, the translocation t(11;19) (q21;p13) has been proposed as the most frequent alteration viewed, found in approximately 60% of cases [40,41]. In our results, we speculated that some genes placed at these chromosomes have their expression altered and may cause changes in the normal function of cells. The expression of TNF-α is encoded by the homonymous gene located on chromosome 6, in region 6p21.33 [42], and it was observed that this chromosome showed numerical abnormalities of monosomy, indicating a lost gene copy, and thereby suggesting less active participation of TNF-α in the tumorigenesis of MEC. These findings corroborate the function assigned to this cytokine to promote programmed cell death, as well as being consistent with its established down-expression in a carcinoma, as it would facilitate tumor development.

In order to verify the probable correlation between MT-2A and the GF and MMPs studied, an siRNA assay was carried out. We observed that, after the silencing of MT2A, a reduction in the expression of TGF-α occurred, indicating a link between these proteins, possibly because MT is involved in the activation process of this GF in MEC. TGF-α, when activated, can promote differentiation and cell growth in head and neck cancers, and thus is considered a pro-mitotic agent in salivary gland neoplasms [13]. An inverse effect was visualized for TNF-α, whose expression increased after MT2A silencing. Hypothetically, the action of TNF-α may be located in the cell signaling pathway, at a stage prior to the action of MT, and the alteration caused by siRNA may have somehow stimulated its expression to compensate for the inhibition of MT. Moreover, we think that as MT was silenced in the cell line, TNF-α, which has the opposite function, became more expressed, and so, probably, its pro-apoptotic action would prevail in the MEC cells.

The siRNA results showed that MT2A silencing did not alter MMP-2 expression. However, MT2A depletion decreased MMP-9 protein levels, indicating a possible correlation between MT and MMP-9. Additionally, discreet expression of TNF-α was observed in the cell group not exposed to siRNA, confirming the cytogenetic findings, which described weak participation of this pro-apoptotic protein once the cancer was already established. Metallothionein acts as a zinc storage protein, and MMPs are proteases dependent on zinc for their function, so the reduction in MMP-9 expression may be explained by MT2A silencing. Our findings also confirm another result, a positive correlation between MT-2A and MMP-9 [12], suggesting the associated participation of these proteins in tumor invasion.

In synthesis, zinc stored by MT could be released to activate transcription factors, which are required for cell proliferation stimulated by TGF-α. MT stores the element required for MMP’s activation. Thus, the release of zinc and its binding to MMPs enables their capacity for matrix extracellular degradation. The anti-apoptotic role of MT in some cancers is well known, so we hypothesized that this action could be related to the possible inactivation of a cytokine involved in apoptosis, like TNF-α.

Moreover, the entire metallothionein pathway has not been established, but its expression and induction have been associated with protection against DNA damage, oxidative stress and apoptosis. Although the cytosolic protein is located in resting cells, it can be transported to the nucleus area to participate in cell proliferation and the differentiation process in a several types of human tumors [6]. 

Our results suggest that MT has an important role in the invasion of cells by MEC, probably by influencing the expression of proteins directly involved in this process, such as TGF-α, TNF-α and MMP-9.

## Figures and Tables

**Figure 1 cells-09-00157-f001:**
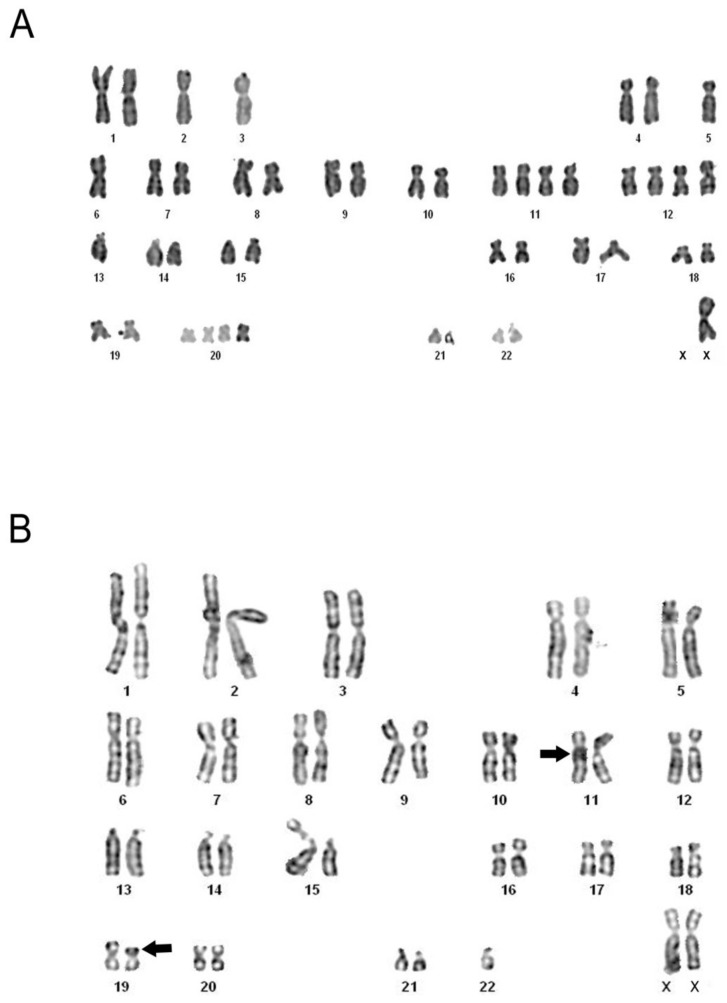
Metaphases from the MEC cell line. G-banded karyotypes revealing various numerical abnormalities of monosomy and tetrasomy (**A**), and the specific translocation of MEC, t(11;19) (q21;p13), indicated by arrows (**B**).

**Figure 2 cells-09-00157-f002:**
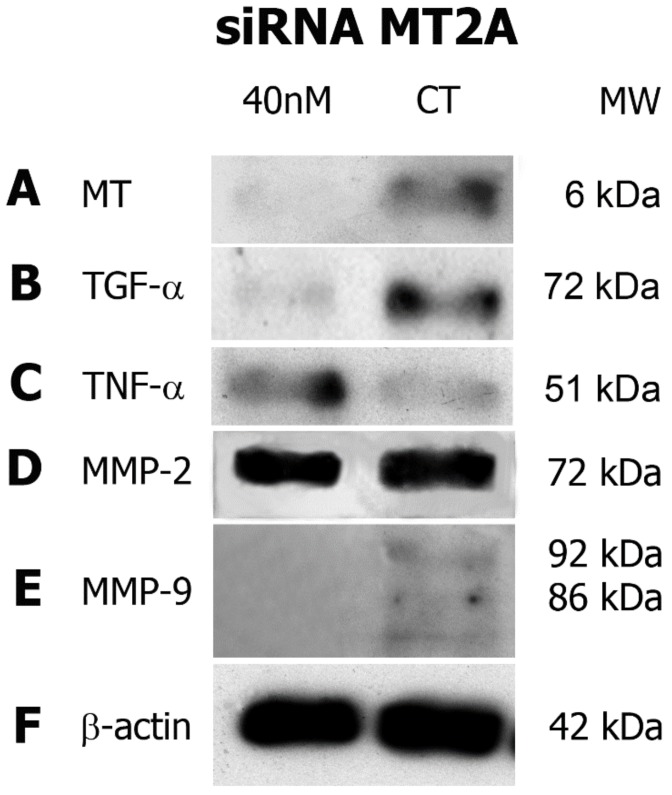
siRNA assay. The experiment promoted a decrease in metallothionein (MT) expression, when compared to the scrambled control (**A**). Similar to MT, the expression of TGF-α was reduced in comparison with the control (**B**). An increase in TNF-α expression was visualized after MT2A gene silencing (**C**). No alteration in MMP-2 expression was found (**D**). Bands of inactive and active MMP-9, with molecular weights of about 92 and 86 kDa, respectively, demonstrated reduced expression after siRNA (**E**). β-Actin internal control presented bands with similar sizes, indicating the correct loading of samples (**D**). nM: nanomolar; CT: control; mW: molecular weight; kDa: kilodaltons.

**Figure 3 cells-09-00157-f003:**
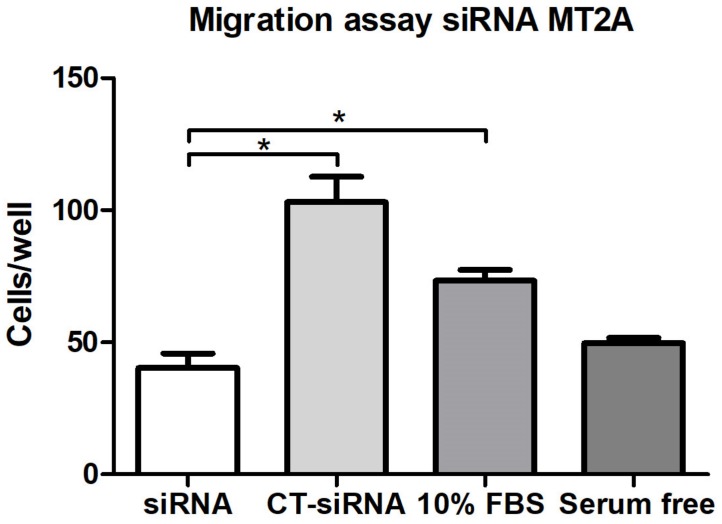
Cell migration assay. A statistically significant difference was observed between the siRNA group and the siRNA control group, as well as between the siRNA group and the positive control (*p* < 0.05). Statistical testing: Mann–Whitney.

**Figure 4 cells-09-00157-f004:**
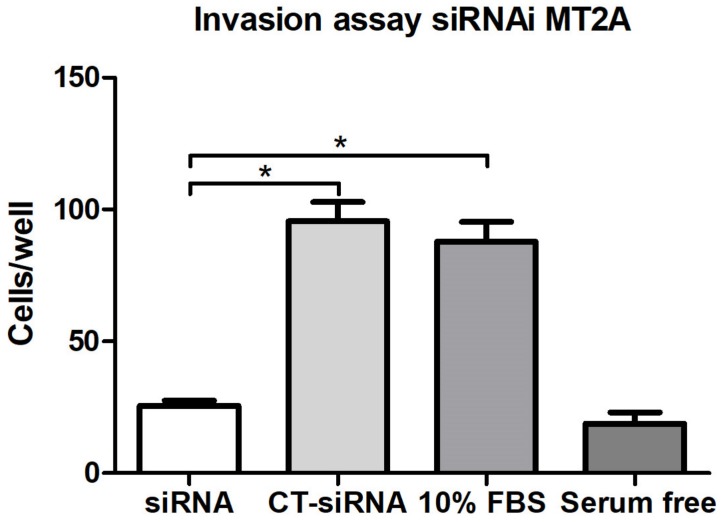
Cell invasion assay. Statistically, a significant difference was observed between the siRNA groups and the siRNA control group, as well as between the siRNA group and the positive control (*p* < 0.05). Statistical testing: Mann–Whitney.

**Table 1 cells-09-00157-t001:** Reads mapped for human salivary gland (HSG) and mucoepidermoid carcinoma (MEC) cell lines.

Gene	Protein	HSG	MEC
*MT2A*	MT-2A	315	3789
*MMP2*	MMP-2	294	63,687
*TNFA*	TNF-α	1	0
*MMP9*	MMP-9	16	2
*KRT7*	CK-7	59,177	569
*KRT8*	CK-8	15,902	9
*KRT14*	CK-14	24	3
*KRT19*	CK-19	24,699	4
*TGFa*	TGF-α	882	89
*VIM*	Vimentin	1	93.082
*FN1*	Fibronectin	934	563,116
*ACTA2*	α-Smooth muscle actin	4526	316,029

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
