# Peer review of "Metallothionein Expression and its Influence on the In Vitro Biological Behavior of Mucoepidermoid Carcinoma"

_cells, 2020, doi:10.3390/cells9010157_

Round 1
Reviewer 1 Report
Cells review
Overall, I think an interesting and important study. Unclear about the pathway from metallothionine to tumorigenesis and invasion. Can this be explained by motility alone (e.g. siRNA to MMP-9). Could use at least one independent siRNA.
Minor weaknesses- grammatical changes needed throughout, e.g.
Line 56. Perhaps “positive feedback”
Line 65 change “mucoepidermoid carcinoma” to MEC (it was already abbreviated
Line 81 “This is probably explained by TNF-α role, which is contrary to tumor progression [15]” needs better grammar
Please check manuscript for grammar throughout.
Major weakness
Line 164 “40 nM siRNA targeting MT2A (Life Technologies, New York, USA), following the manufacturer’s instructions”- why only one siRNA? Is sequence known? Ideally should be three, or at least two siRNAs plus one scrambled.
Figure 2 and 3 re-use the same Figures- maybe could be placed in the same Figure.
Model for how MT affects the expression of each of these markers.
More of discussion of role of metallothionine
No mechanism is provided for the metallothionein pathway- authors mention the importance of zinc in proteases, but this is not a tight connection with metallothionein
(maybe prevent oxidation by binding metal?),
Also, the roles of different types of metallothionine.
Line 218-219 “Transcriptomic analysis revealed that the MT2A gene showed a significantly higher number of reads mapped in the MEC cell line (3789) compared to the HSG line (315). These results reinforce the pronounced expression of MT2A in tumor cells as a relevant prognostic marker.” How do you know that this is not just a marker of mesenchymal cells vs. HSG cell line which is more epithelial?
In Figure 3, there is a reduction in MMP-9 but not MMP-2, however the immunoblot, as well as the previous RNA data shows very little MMP-9 to begin with? Also, MMP-2 looks unchanged. So how would this be enough to alter invasion? Could you explain this all by increased migration? Did the authors examine the secreted active forms of MMP2, 9 by concentrating the medium and running western?
Author Response
Revisor 1
Cells review
Overall, I think an interesting and important study. Unclear about the pathway from metallothionine to tumorigenesis and invasion. Can this be explained by motility alone (e.g. siRNA to MMP-9). Could use at least one independent siRNA.
Minor weaknesses- grammatical changes needed throughout, e.g.
Line 56. Perhaps “positive feedback” Reply: It has been corrected and it now reads: “Growth factors also promote the secretion of MMPs, providing a positive feedback mechanism favorable to tumor invasion”
Line 65 change “mucoepidermoid carcinoma” to MEC (it was already abbreviated) Reply: It has been corrected and it now reads: “However, little is known about its role in MEC invasiveness”
Line 81 “This is probably explained by TNF-α role, which is contrary to tumor progression [15]” needs better grammar Reply: It has been corrected and it now reads: “This could be explained for TNF-α role, which act contrary to tumor development”
Please check manuscript for grammar throughout.
Major weakness
Line 164 “40 nM siRNA targeting MT2A (Life Technologies, New York, USA), following the manufacturer’s instructions”- why only one siRNA? Is sequence known? Ideally should be three, or at least two siRNAs plus one scrambled.
Reply: This is a validated sequence, which provides an adequate level of silencing to MT2A gen.
A new period has been included (line 161) and it now reads:
Small interfering RNAs against MT2A (siRNA s226679) were manufactured by Life Technologies: 5’-CAAAUGCACUUCGUGCAAGtt-3’ and 5’-CUUGCACGAAGUGCAUUUGca-3’. MEC cells (2 × 105) were cultured in six-well plates in DMEM-F12 with 10% FBS and without antibiotic–antimycotic solution to 60–80% confluence. Cells were incubated with a complex formed by transfection medium (Optimen, Invitrogen Molecular Probes, Carlsbad, CA, USA), Lipofectamine 2000 transfection reagent (Invitrogen®) and 40 nM siRNA targeting MT2A (Life Technologies, New York, USA), following the manufacturer’s instructions. A 40-nM siRNA scrambled sequence (Santa Cruz proprietary target sequence) was used as control. Transfection efficiency was demonstrated by western blot.
Figure 2 and 3 re-use the same Figures- maybe could be placed in the same Figure Reply: Figure 2 and 3 were placed in the same figure.
Model for how MT affects the expression of each of these markers. Reply: It has been included a new paragraph (line 372), which now states:
In syntesis, zinc stored by MT could be released to activate transcriptions factors, which are required to cell proliferation stimulated by TGF-α. MT stores the element required for MMP’s activation. Thus, release of zinc and its binding to MMPs enables their capacity of matrix extracellular degradation. Anti-apoptotic role of MT in some cancer is well known, so we hypothesized that this action could be related to a possible inactivation of a cytokine involved in apoptosis like a TNF- -α.
More of discussion of role of metallothionine No mechanism is provided for the metallothionein pathway- authors mention the importance of zinc in proteases, but this is not a tight connection with metallothionein (maybe prevent oxidation by binding metal?).
Reply: We addressed this point by including new statements in the manuscript (line 372):
Moreover, the entire metallothionein pathway has not been established, but its expression and induction also have been associated with protection against DNA damage, oxidative stress and apoptosis. Although be a cytosolic protein in resting cells, it can be transported to nucleus area to participate of cell proliferation and differentiation process in a several types of human tumors [6].
Also, the roles of different types of metallothionine.
Reply: It has been included a new paragraph (line 330) and it now reads:
Four isoforms has been studied, in which MT-3 and MT-4 are expressed in the brain and in the squamous epithelia, respectively. MT-1 and MT-2 are present in most organs, but their role has not been elucidated [12]. A specific isoform of this protein, MT-2A, has been widely studied as a poor prognostic factor in breast cancer, with its presence associated with a more aggressive tumor behavior [12].
Line 218-219 “Transcriptomic analysis revealed that the MT2A gene showed a significantly higher number of reads mapped in the MEC cell line (3789) compared to the HSG line (315). These results reinforce the pronounced expression of MT2A in tumor cells as a relevant prognostic marker.” How do you know that this is not just a marker of mesenchymal cells vs. HSG cell line which is more epithelial? Reply: MT2A is considered a metallothionein isoform strongly associated to less differentiated tumors, useful as a poor marker of cancer. Moreover, new statements were included in the discussion (line 293):
Besides, the higher expression of vimentin and α-smooth muscle actin in the MEC cell line, compared to the HSG line, possibly demonstrate presence of myoepithelial cells, in which are part of cancer-associated fibroblasts (CAFs) and was already visualized in MEC [41]. Moreover, when MT is expressed in CAFs, the tumor invasiveness capacity is increased in uterine cervical carcinoma [42].
Two new references were added to list.
In Figure 3, there is a reduction in MMP-9 but not MMP-2, however the immunoblot, as well as the previous RNA data shows very little MMP-9 to begin with? Also, MMP-2 looks unchanged. So how would this be enough to alter invasion? Could you explain this all by increased migration? Did the authors examine the secreted active forms of MMP2, 9 by concentrating the medium and running western?
Reply: We consider that presence of Matrigel, which corresponds to basement membrane in vitro, could be a stimulus to MEC cells to secret more MMP-9 and have a proteolytic action, reaching lower chamber wells. In addition, there are different times intervals to MMP9 secretion between the methods. For example, to migration and invasion assays, this period is of 24 and 48 hours, respectively, which not happens in transcriptomics analysis.

Reviewer 2 Report
In their manuscript entitled “Metallothionein expression and its influence on the in vitro biological behavior of mucoepidermoid carcinoma” the authors aimed to characterize a cell line derived from mucoepidermoid carcinoma (MEC) and to correlate metallothionein expression with the expression of transforming growth factor-α, tumor necrosis factor-α, and matrix metalloproteinases. They evaluate the role of metallothionein in the MEC cells migration and invasion.
I found this manuscript interesting, although the role of matallothionnein in carcinogenesis has already been widely studied. I have some comments for the authors as detailed below.
In the Materials and Methods section - authors should include information about the source of the cells used in the study, where they were purchased (name of the cell line) or how they were isolated (if they originate from primary culture). In the Materials and Methods section - the numbers of cells used in the experiments should be written using superscripts, because this was probably leveled out when editing the text. The Results section: Page 8, line 261 - invalid subtitle in relation to the content presented below (repetition of the subtitle - page 6, line 239) How do the authors explain the lower expression of TGF-α and MMP-9 (taking into account the role of these factors) in MEC than in HSG cell line or the small difference in TNF-α expression between these lines (Table 1)? The Discussion section, page 9, lines 328-330 - the authors wrote: " A specific isoform of this protein, MT-2A, has been widely studied as a poor prognostic factor in salivary gland neoplasms, with its presence associated with a more aggressive tumor behavior [12]." - this sentence requires changes or adding other references, as the indicated one refers to breast cancer.Author Response
Reviewer 2
Comments and Suggestions for Authors
In their manuscript entitled “Metallothionein expression and its influence on the in vitro biological behavior of mucoepidermoid carcinoma” the authors aimed to characterize a cell line derived from mucoepidermoid carcinoma (MEC) and to correlate metallothionein expression with the expression of transforming growth factor-α, tumor necrosis factor-α, and matrix metalloproteinases. They evaluate the role of metallothionein in the MEC cells migration and invasion.
I found this manuscript interesting, although the role of matallothionnein in carcinogenesis has already been widely studied. I have some comments for the authors as detailed below.
In the Materials and Methods section - authors should include information about the source of the cells used in the study, where they were purchased (name of the cell line) or how they were isolated (if they originate from primary culture).
Reply: It has been corrected and it now reads:
“A cell line derived from primary culture of a MEC[…]”
In the Materials and Methods section - the numbers of cells used in the experiments should be written using superscripts, because this was probably leveled out when editing the text.
Reply: It has been corrected and the numbers of cells used in the experiments are written using superscripts.
The Results section: Page 8, line 261 - invalid subtitle in relation to the content presented below (repetition of the subtitle - page 6, line 239)
Reply: It has been corrected and it now reads:
“3.6. MT2A silencing decreases migratory and invasive activity in MEC cells”
How do the authors explain the lower expression of TGF-α and MMP-9 (taking into account the role of these factors) in MEC than in HSG cell line or the small difference in TNF-α expression between these lines (Table 1)?
Reply: Even with higher expression of TGF-α e MMP-9 in HSG than in MEC cell line, it could be not enough to become actives and do its functions because they need the zinc substrate stored by MT-2A and it has low expression in HSG cell line. Besides, MMP-9 could be more down regulated by the tissue inhibitors of metalloproteinase’s (TIMPs) in HSG cells.
The Discussion section, page 9, lines 328-330 - the authors wrote:
“A specific isoform of this protein, MT-2A, has been widely studied as a poor prognostic factor in salivary gland neoplasms, with its presence associated with a more aggressive tumor behavior [12]." - this sentence requires changes or adding other references, as the indicated one refers to breast cancer.
Reply: It has been corrected and it now reads:
"A specific isoform of this protein, MT-2A, has been widely studied as a poor prognostic factor in breast cancer, with its presence associated with a more aggressive tumor behavior [12]."

Round 2
Reviewer 1 Report
As previously stated, the authors are using only one siRNA, which is considered to be a concern because of potential off-target effects for all siRNAs. The siRNAs given in the methods are just listing both strands of one siRNA.
The manuscript could be still be improved with proofreading for grammar.
Author Response
UNIVERSIDADE FEDERAL DO PARÁ
INSTITUTO DE CIÊNCIAS DA SAÚDE
FACULDADE DE ODONTOLOGIA
Belém, 4th December, 2019
REVISION NOTES
Reviewer 1
A new edit of all text has been made (certificate attached)
As previously stated, the authors are using only one siRNA, which is considered to be a concern because of potential off-target effects for all siRNAs.
The siRNAs given in the methods are just listing both strands of one siRNA.
Reply: We now understand the questioned point:
We bought four siRNAs from Thermo (Life), s226631 Cat. # 4392420, s194629 Cat. # 4392420, s226679 Cat. # 4392420 and s194630 Cat. # 4392420.
For this work, we used siRNA s226631 and s226679 that act in different regions of the target gene.
A new period has been included (lines 161-163) and it now reads:
Small interfering RNAs against MT2A (siRNA s226631 and s226679, Life Technologies) were used. This is a Silencer Select (Thermo®), chemically modified siRNA, with a proprietary chemical modification that reduces overall off-target effects by up to 90% without compromising potency.
We certify that the following article
Metallothionein expression and its influence on the in vitro biological behavior of mucoepidermoid carcinoma
João de Jesus Pinheiro
has undergone English language editing by MDPI. The text has been checked for correct use of grammar and common technical terms, and edited to a level suitable for reporting research in a scholarly journal.
MDPI uses experienced, native English speaking editors. Full details of the editing service can be found at
https://www.mdpi.com/authors/english.
Basel, December 2019
Martyn Rittman, Ph.D. English Editing Manager englishediting@mdpi.com
https://www.mdpi.com/authors/english
